# Efficacy and Durability of Dolutegravir- or Darunavir-Based Regimens in ART-Naïve AIDS- or Late-Presenting HIV-Infected Patients

**DOI:** 10.3390/v15051123

**Published:** 2023-05-08

**Authors:** Massimiliano Fabbiani, Melissa Masini, Barbara Rossetti, Arturo Ciccullo, Vanni Borghi, Filippo Lagi, Amedeo Capetti, Manuela Colafigli, Francesca Panza, Gianmaria Baldin, Cristina Mussini, Gaetana Sterrantino, Damiano Farinacci, Francesca Montagnani, Mario Tumbarello, Simona Di Giambenedetto

**Affiliations:** 1UOC Malattie Infettive e Tropicali, Azienda Ospedaliero-Universitaria Senese, 53100 Siena, Italy; 2UOC Malattie Infettive, Azienda USL Toscana Sud Est, PO San Donato, 52100 Arezzo, Italy; 3Division of Infectious Diseases, AUSL Toscana Sud Est, Grosseto Hospital, 58100 Grosseto, Italy; 4Dipartimento di Sicurezza e Bioetica Sezione Malattie Infettive, Università Cattolica del Sacro Cuore, 00168 Rome, Italy; 5UOC Malattie Infettive, Fondazione Policlinico Universitario A. Gemelli IRCCS, 00168 Rome, Italy; 6Clinica Malattie Infettive e Tropicali, Azienda Ospedaliero Universitaria di Modena, 41100 Modena, Italy; 7SOD Malattie Infettive e Tropicali, Azienda Ospedaliero-Universitaria Careggi, 50134 Firenze, Italy; 8Divisione di Malattie Infettive, Dipartimento di Malattie Infettive, Ospedale Universitario Luigi Sacco, 20157 Milano, Italy; 9Unità di Dermatologia Infettiva e Allergologia, Istituto S. Gallicano IRCCS, 00144 Rome, Italy; 10Dipartimento di Biotecnologie Mediche, Università degli Studi di Siena, 53100 Siena, Italy; 11Department of Experimental and Clinical Medicine, University of Florence, 50134 Florence, Italy

**Keywords:** HIV-1, antiretroviral therapy, integrase inhibitors, protease inhibitors, advanced naïve, virological failure, treatment discontinuation

## Abstract

Background: Since limited data are available, we aimed to compare the efficacy and durability of dolutegravir and darunavir in advanced naïve patients. Methods: Retrospective multicenter study including AIDS- or late-presenting (def. CD4 ≤ 200/µL) HIV-infected patients starting dolutegravir or ritonavir/cobicistat-boosted darunavir+2NRTIs. Patients were followed from the date of first-line therapy initiation (baseline, BL) to the discontinuation of darunavir or dolutegravir, or for a maximum of 36 months of follow-up. Results: Overall 308 patients (79.2% males, median age 43 years, 40.3% AIDS-presenters, median CD4 66 cells/µL) were enrolled; 181 (58.8%) and 127 (41.2%) were treated with dolutegravir and darunavir, respectively. Incidence of treatment discontinuation (TD), virological failure (VF, defined as a single HIV-RNA > 1000 cp/mL or two consecutive HIV-RNA > 50 cp/mL after 6 months of therapy or after virological suppression had been achieved), treatment failure (the first of TD or VF), and optimal immunological recovery (defined as CD4 ≥ 500/µL + CD4 ≥ 30% + CD4/CD8 ≥ 1) were 21.9, 5.2, 25.6 and 1.4 per 100 person-years of follow-up, respectively, without significant differences between dolutegravir and darunavir (*p* > 0.05 for all outcomes). However, a higher estimated probability of TD for central nervous system (CNS) toxicity (at 36 months: 11.7% vs. 0%, *p* = 0.002) was observed for dolutegravir, whereas darunavir showed a higher probability of TD for simplification (at 36 months: 21.3% vs. 5.7%, *p* = 0.046). Conclusions: Dolutegravir and darunavir showed similar efficacy in AIDS- and late-presenting patients. A higher risk of TD due to CNS toxicity was observed with dolutegravir, and a higher probability of treatment simplification with darunavir.

## 1. Introduction

Current international guidelines recommend starting antiretroviral therapy (ART) in all HIV-infected patients, regardless of their CD4 cell count [1,2], as early ART initiation has found to be associated with reduced morbidity and mortality in several studies [3,4]. In particular, early ART proved beneficial in reducing the incidence of serious AIDS-related events, serious non-AIDS-related events, and death from any cause. However, a substantial proportion of patients (i.e., reaching up to 50% also in high-income countries) are still being diagnosed with HIV at low CD4 counts or at the time of AIDS occurrence [5], suggesting that efforts to ensure a timely diagnosis of HIV are still needed. Late diagnosis and treatment are associated with a higher risk of virological failure, incomplete CD4 recovery, disease progression and death [6,7,8,9]. Although many innovative approaches have been undertaken to increase and target HIV testing, and an increase in the number of tests performed throughout Europe has been reported, there has been no decline in the proportion of patients diagnosed with advanced HIV infection since 2010 in Europe [5]. Several reasons have been postulated for this epidemiological phenomenon, such as different testing strategies, difficulties in identifying and actually reaching populations at higher risk of HIV infection, a low self-perceived risk of infection, limited knowledge of the disease, and barriers to testing and treatment.

Although late presentation is common, the optimal therapeutic strategy in advanced disease is still being debated. Theoretically, starting treatment soon after diagnosis should be a priority in late presenters, except for those patients diagnosed with certain opportunistic infections for which deferred ART is recommended to avoid the immune reconstitution inflammatory syndrome (IRIS) [10,11]. It has been demonstrated that early initiation of ART is associated with a substantial reduction in mortality for most patients with opportunistic infections or other AIDS-defining events [12]. However, as studies on the efficacy of ART in patients with advanced disease are scant, it has not yet been clearly defined which ART regimen leads to a better outcome. Ideally, the optimal antiretroviral regimen for late presenters should have high efficacy and a high genetic barrier; thus, treatment may be started early, before obtaining genotypic resistance test results, which is time-consuming in most clinical settings. The latest generation of integrase inhibitors (InSTI) (i.e., dolutegravir and bictegravir) have all of these characteristics, as well as high tolerability and a low potential for drug–drug interactions [13]; therefore, current guidelines recommend them as the preferred anchor drugs in first-line regimens [1,2]. However, very few data are available for InSTI-based regimens in patients with a low CD4 count, as these subjects are poorly represented in clinical trials [14,15,16], and few observational studies are available [17,18]. Moreover, theoretically, the faster virological decay observed with InSTI-based regimens [19,20,21,22] could potentially translate into rapid immunological recovery [23], and could be associated with the development of IRIS, which could be deleterious in this setting. Although protease inhibitors (PI) show less tolerability, they may have some characteristics (i.e., high genetic barrier, slower viral decay despite good long term efficacy, recent availability in a single-tablet regimen) [24] that make them an attractive alternative option in late presenters. Darunavir is a drug with a higher genetic barrier, and it has demonstrated a higher tolerability than older PIs [21]. Trials exploring the efficacy of darunavir as a first-line treatment also included patients with low CD4 counts and AIDS-defining events, and they also reported good results in the long term [25]. Moreover, experience in the use of darunavir to treat advanced HIV infection in a real-life setting has been established. However, few studies have compared last-generation InSTI and PI in this setting to determine which option is associated with better outcomes.

The aim of our study was to describe and compare the long-term efficacy and durability of dolutegravir- and darunavir-based regimens in ART-naïve AIDS and late-presenting HIV-infected patients in clinical practice.

## 2. Materials and Methods

### 2.1. Patients and Follow-Up

We performed a retrospective multicenter study (six reference centers in central and northern Italy). We analyzed ART-naïve adult HIV-1 infected, AIDS- or late-presenting (defined as having a baseline CD4 count ≤ 200/µL) patients, who were administered first-line therapy with dolutegravir 50 mg or ritonavir/cobicistat-boosted darunavir 800 mg once daily + two nucleoside reverse transcriptase inhibitors (NRTI) from January 2009 to June 2019. All patients started treatment in accordance with national or international guidelines and at the clinical judgement of the caring physicians, during routine clinical practice. The exclusion criteria were as follows: age < 18 years, starting treatment during acute HIV infection (defined as having a Western blot demonstrating HIV infection in Fiebig stages I–V, or as having a negative HIV test in the last six months) [8,26,27], starting regimens that included more than three drugs, and not having clinical or laboratory follow-up data.

Patients were followed from the date of first-line ART initiation (baseline, BL) to the discontinuation of darunavir or dolutegravir, the last available visit, death, loss to follow-up or a maximum of 36 months of follow-up (whichever occurred first).

Baseline characteristics and laboratory data were retrieved from electronic databases or chart review. All patients signed an informed consent for the use of their clinical and laboratory data in aggregated and anonymous form, and were aware that the databases could be used to produce observational studies. The data collection procedure was presented to the Ethics Committees of the centres. Access to the database and to data analyses is regulated by local institutional Ethics Committees and conforms to Italian and European privacy legislations.

### 2.2. Main Endpoints

Four main endpoints were evaluated: (i) the time to treatment discontinuation (TD) of dolutegravir or darunavir, overall and according to different reasons (note that the switch of backbone or that from ritonavir to cobicistat were not considered as TD); (ii) the time to virological failure (VF), defined as having HIV-RNA > 50 copies/mL in two consecutive determinations after 6 months from ART initiation or, after the achievement of virological suppression, having a rebound above 50 copies/mL in two consecutive determinations or >1000 copies/mL in a single determination [8,23,28]; (iii) the time to treatment failure (TF), a composite outcome defined as the first of VF or TD; (iv) the time to optimal immunological recovery (OIR), defined as achievement of CD4 ≥ 500/mmc + CD4 ≥ 30% + CD4/CD8 ratio ≥ 1 [8,23].

### 2.3. Statistical Analysis

Descriptive statistics [number, proportion, median, interquartile range (IQR), 95% confidence intervals (CI)] were used to describe the patients’ baseline characteristics. Categorical variables were compared between groups using the Chi-square test or Fisher’s exact test, as appropriate. Continuous variables were compared using the non-parametric Kruskal–Wallis and Mann–Whitney U test. Kaplan–Meier curves and Cox regression analyses were used to estimate incidence and predictors of time to TD (overall and according to different reasons), VF, TF and OIR. Variables showing a *p* value < 0.100 in the univariate analysis, together with treatment arm (darunavir versus dolutegravir, i.e., the variable of interest), were then investigated in a multivariate model. Only *p* values < 0.05 were considered significant. All analyses were performed using the SPSS version 18.0 software package (SPSS Inc., Chicago, IL, USA).

## 3. Results

### 3.1. Population Characteristics

A total of 308 patients were included; their main characteristics are reported in Table 1. Overall, 244 (79.2%) were males with a median age of 42.9 years (IQR 35.1–51.2). An AIDS-defining event had been diagnosed in 124 (40.3%) subjects at the time of ART initiation (details reported in Appendix A); of these, 46 (37.1%) patients had experienced ≥1 AIDS event before ART initiation. Considering all AIDS events (n = 187), the most common was pneumocystis jirovecii pneumonia (n = 52, 28.8%), followed by oesophageal candidiadis (n = 31, 16.6%), cytomegalovirus diseases (n = 8, 15%), Kaposi sarcoma (n = 18, 9.6%) and wasting syndrome (n = 15, 8%). At baseline, the median HIV-1 RNA was 5.29 log_10_ copies/mL (IQR 4.92–5.78), the median CD4 cell count was 66 cell/μL (IQR 25–122), and the median CD4:CD8 ratio was 0.10 (0.05–0.20).

Overall, 181 (58.7%) and 127 (41.3%) individuals started first-line ART regimens that included dolutegravir or darunavir, respectively. The baseline characteristics of the patients who started dolutegravir or darunavir were quite homogenous. However, in the darunavir group, there was a higher proportion of HCV-coinfected subjects (6.3% versus 3.3%, *p* = 0.021), and of tenofovir disoproxil fumarate (TDF) or tenofovir alafenamide (TAF) plus emtricitabine (FTC) use (85.8% versus 72.4%, *p* = 0.002).

### 3.2. Treatment Discontinuation

During a median follow-up of 17.2 (IQR 5.7–34.4) months, 113 (36.7%) patients underwent TD (n = 49/181, 27.1% dolutegravir; n = 64/127, 50.4% darunavir). The incidence of TD was 18.2 per 100 person-years of follow-up (PYFU) for dolutegravir, and 21.51 per 100 PYFU for darunavir. Overall, Kaplan–Meier curves did not show a significant difference in TD between dolutegravir and darunavir (log rank *p* = 0.147) (Figure 1).

However, higher rates of TD were observed for dolutegravir in the first 12 months of treatment; the estimated TD was 25.5% with dolutegravir versus 17.6% with darunavir at 12 months (log rank *p* = 0.057), when follow-up was censored at one year. Conversely, higher rates of TD were observed for darunavir after the first 12 months of treatment; the estimated TD was 9.1% with dolutegravir versus 40.6% with darunavir at 36 months (log rank *p* < 0.001), when only patients still on first-line treatment after 12 months were considered.

Upon multivariable Cox regression analyses, AIDS was the only predictor of TD [adjusted hazard ratio (aHR) 1.49, 95% CI 1.00–2.22, *p* = 0.051] after adjusting for treatment arm (darunavir versus dolutegravir aHR 1.24, 95% CI 0.85–1.82, *p* = 0.264) and CD4+ count (when compared to CD4 > 100 cell/μL: CD4 < 50 cell/μL aHR 1.22, 95% CI 0.76–1.96, *p* = 0.420; CD4 50–100 cell/μL aHR 1.42, 95% CI 0.86–2.35, *p* = 0.169).

The specific reasons for treatment discontinuation are reported in Table 2.

Dolutegravir TD was primarily due to toxicity [13.3% (n = 24), of which 5.5% (n = 10) for central nervous system (CNS) toxicity] followed by simplification (n = 9, 5%). Darunavir TD was primarily due to simplification (n = 26, 20.5%) followed by toxicity (n = 20, 15.7%, with no cases of CNS toxicity) and drug–drug interactions (n = 6, 4.7%). No cases of IRIS were reported for either drug.

No significant differences were observed between dolutegravir and darunavir regarding the incidence of TD for overall toxicity (dolutegravir 18.2 per 100 PYFU versus darunavir 21.5 per 100 PYFU; the estimated proportion of TD for toxicity at 36 months was 17.3% for dolutegravir and 19.4% for darunavir, log rank *p* = 0.791) (Figure 2A).

However, the incidence of TD for CNS toxicity (TD-CNS) was significantly higher for dolutegravir (3.7 per 100 PYFU) than for darunavir (0 per 100 PYFU) (Figure 2B); the estimated proportion of TD-CNS at 36 months was 8.7% for dolutegravir and 0% for darunavir (log rank *p* = 0.004).

The incidence of TD for simplification (TD-S) for dolutegravir was 3.4 per 100 PYFU versus 8.7 per 100 PYFU for darunavir, with an estimated 7.7% proportion of TD at 36 months for dolutegravir and 22.1% for darunavir (log rank *p* = 0.009) (Figure 2C). Darunavir also showed a higher risk of TD-S in the multivariate analysis (aHR 2.47, 95% CI 1.14–5.38, *p* = 0.022) after adjusting for age (aHR 0.61 per 10 years increase, 95% CI 0.43–0.86, *p* = 0.005) and backbone (when compared to TDF or TAF + FTC: abacavir + lamivudine aHR 0.56, 95% CI 0.17–1.91, *p* = 0.356).

The incidence of TD due to drug–drug interactions (TD-DDI) was significantly lower for dolutegravir (0.4 per 100 PYFU) than darunavir (2.0 per 100 PYFU) (Figure 2D). The estimated proportion of TD-DDI at 36 months was 0.6% for dolutegravir and 6.5% for darunavir (log rank *p* = 0.041).

### 3.3. Virological Failure

Overall, 21 (6.8%) patients experienced virological failure (VF) (n = 9/181, 5% with dolutegravir and n = 12/127, 9.4% with darunavir). The incidence of virological failure was 3.75 per 100 PYFU for dolutegravir and 4.6 per 100 PYFU for darunavir. The estimated proportion of VF at 12 months was 4.7% for dolutegravir and 7.8% for darunavir, and at 36 months it was 9.7% for dolutegravir and 12.4% for darunavir (log rank *p* = 0.589) (Figure 3A). A Cox multivariate analysis indicated that darunavir did not demonstrate a significantly higher risk of VF (aHR 1.22, 95% CI 0.47–3.16, *p* = 0.687) than dolutegravir after adjusting for CD4 percentage and CD4/CD8 ratio at baseline.

### 3.4. Treatment Failure

Overall, 124 (40.3%) patients experienced treatment failure (TF) (n = 54/181, 29.8% with dolutegravir and n = 70/127, 55.1% with darunavir). The incidence of TF was 20.7 per 100 PYFU in the dolutegravir arm and 25.6 per 100 PYFU in the darunavir arm (*p* = 0.105). The estimated proportion of TF at 12 months was 28.4% for dolutegravir and 22.5% for darunavir, and at 36 months it was 36.9% for dolutegravir and 56.4% for darunavir (Figure 3B). A Cox multivariate analysis indicated that darunavir did not lead to a significantly higher risk of TF (aHR 1.18, 95% CI 0.79–1.77, *p* = 0.412) when compared to dolutegravir, after adjusting for AIDS-presenting, absolute CD4 count, CD4 percentage and CD4/CD8 ratio at baseline.

### 3.5. Optimal Immunological Recovery

Overall, 8 (2.6%) patients showed optimal immunological recovery (OIR) (n = 6/181, 3.3% with dolutegravir and n = 2/127, 1.6% with darunavir). The incidence of OIR was 2.25 per 100 PYFU in the dolutegravir arm and 0.82 per 100 PYFU in the darunavir arm (*p* = 0.245). The estimated proportion of OIR at 12 months was 1.7% for dolutegravir and 1.1% for darunavir, and at 36 months it was 5.1% for dolutegravir and 1.1% for darunavir (Figure 3C). In a Cox multivariate analysis, darunavir did not show a significantly lower probability of OIR (aHR 0.61, 95% CI 0.11–3.53, *p* = 0.584) when compared to dolutegravir, after adjusting for absolute CD4 count, CD4 percentage and CD4/CD8 ratio at baseline.

## 4. Discussion

Individuals with a low CD4 count or with concomitant AIDS-defining illnesses constitute a relevant proportion of patients at the time of HIV diagnosis [5]; however, they are underrepresented in clinical trials exploring the efficacy of first-line ART. In the FLAMINGO trial, dolutegravir demonstrated superior efficacy at 48 weeks in treatment of naïve patients, when compared to ritonavir-boosted darunavir [14]. However, in this study, patients with active opportunistic infections or other AIDS-defining diseases were excluded, and only 10% of the patients had baseline CD4 < 200 cells/μL. Populations included in other trials exploring first-line ART have similar characteristics [15,16,29]. Thus, limited data on the efficacy of antiretroviral drugs in late presenters are available from randomized clinical trials. Moreover, few studies comparing PIs and InSTIs have been performed in this difficult-to-treat population in clinical practice, and most have some drawbacks that limit the generalizability of the results to the current treatment options. Indeed, some studies included older PIs (i.e., atazanavir/ritonavir, lopinavir/ritonavir) or a first-generation InSTI (i.e., raltegravir, elvitegravir) [17,18,30], which, as of the last few years, is no longer recommended. Other studies included darunavir or a late-generation InSTI (i.e., dolutegravir and bictegravir), but had a limited follow-up which did not permit adequate investigation of the dynamics of TD in the long term [31,32]. Therefore, additional data are needed to guide the choice of antiretroviral regimens in late presenters.

In our study, we analysed a population of patients with advanced HIV disease, and compared the long-term efficacy and durability of dolutegravir- and darunavir-based regimens, i.e., two drugs with a high genetic barrier that are often recommended in this setting.

Both regimens had high rates of virological efficacy, with an estimated proportion of virological failure of 9.7% and 12.4% at 36 months for dolutegravir and darunavir, respectively. This is in accordance with data also showing that late presenters, a difficult-to-treat population, can achieve high rates of virological success with current antiretroviral regimens [30]. Unfortunately, no data on the emergence of drug resistance mutations were available for our population, so we could not evaluate whether virological failure with the two treatment regimens might have a different impact on future treatment options. However, data from clinical trials suggest that the emergence of resistance upon first-line ART failure is a rare phenomenon [33].

To evaluate immunological recovery, we focused on the OIR index which combines the absolute CD4 count, percentage of CD4 and CD4:CD8 ratio [8,23]. This parameter may better reflect the recovery of immune function in the mid- or long-term, since it combines variables with different biological significance. While the absolute CD4 count is the strongest predictor of disease progression and survival [34,35], it has also been suggested that the CD4 percentage is a marker of disease progression [36]. Moreover, the CD4:CD8 ratio reflects T-cell activation, innate immune activation, and the presence of an immunosenescent T-cell phenotype [37]; indeed, a low CD4:CD8 ratio has been associated with non-AIDS-defining events and mortality [38,39]. In our population, both regimens showed a similar immunological recovery based on the OIR index. However, OIR was infrequently reached with both regimens at the mid term, as it occurred only for 5.1% of patients with dolutegravir and 1.1% with darunavir at 36 months. This suggest that patients with advanced immune deficiency might have reduced or delayed immune recovery, even after reaching a stable virological suppression [39].

Some discrepancies were observed between the two regimens regarding rates of treatment discontinuation. Specifically, dolutegravir showed a higher risk of treatment discontinuation during the first 12 months of treatment, mainly due to CNS toxicity. Conversely, darunavir showed a higher risk of treatment discontinuation after the first 12 months of therapy, mainly due to simplification and drug–drug interactions.

Dolutegravir has been associated with the development of CNS toxicity, especially neuropsychiatric side effects [40]. In some cohorts, rates of discontinuation of this drug due to CNS adverse events were quite high [41,42], especially if dolutegravir was associated with an abacavir-containing backbone [41]. However, the incidence of dolutegravir discontinuation in other cohorts was much lower [43]. These discrepancies suggest that the development of CNS symptoms is a multifactorial process, and that several variables might be associated with this toxicity (e.g., age, gender, comorbidities, concomitant medications, inter-individual variability in drug pharmacokinetics). In our cohort, the incidence of dolutegravir discontinuation due to CNS toxicity was not very high (estimated 8.7% at 36 months), especially compared to what has been observed with other antiretroviral drugs that are known to produce neuropsychiatric side effects (e.g., efavirenz) [44]. However, it was higher than that observed in another Italian cohort, where only 3.4% of patients discontinued dolutegravir because of CNS side effects [43]. In the latter study, most patients started dolutegravir to simplify treatment, and only a minority of subjects were treatment-naïve. It has been suggested that the discontinuation rates for dolutegravir due to CNS toxicity might be higher in treatment-naïve subjects [42]. Since our population included naïve patients with advanced disease, it could be inferred that this difficult-to-treat population could have had more CNS problems at baseline, and this may have affected the tolerability of dolutegravir. Whether long-term exposure to dolutegravir is associated with the development of cognitive deterioration is a matter of debate and should be explored in future studies.

It should also be noted that some reports shows that InSTI might be associated with higher rates of IRIS in advanced patients [45,46]. Since these toxicities usually develop in the short term, they could contribute to the higher rates of treatment discontinuation of dolutegravir in the first 12 months of ART. However, no cases of IRIS were reported in our cohort for either drug, which suggests that this phenomenon is rarely treatment limiting also in advanced naïve patients [17].

Indeed, darunavir has a lower metabolic tolerability and a higher potential for drug–drug interactions when compared to InSTI [47]. The potential for significant drug interactions is high, since this drug has a potent inhibitory effect on cytochrome CYP 450, thus potentially altering the pharmacokinetics of all drugs metabolized by this pathway [48]. Moreover, metabolic toxicities might mainly emerge in the middle and long term, when the need to suppress viral replication has been achieved. This could explain the higher risk of treatment discontinuation of darunavir-based regimens after the first 12 months of treatment. It should also be emphasized that in the last few years, some InSTI-based two-drug regimens have become available, and have also demonstrated good efficacy in the middle and long term [49]. Moreover, their tolerability seems higher when compared to standard three-drug regimens, especially those based on protease inhibitors. The switch to two-drug ART could also explain part of the treatment discontinuation for simplification, especially for regimens based on darunavir. Unfortunately, no data on subsequent ART regimens were available in our dataset.

A strength of our study was the availability of a long-term follow-up, which allowed us to estimate discontinuation rates up to 36 months from first-line therapy initiation. Moreover, it should be emphasized that our definition of TD was specifically related to anchor drugs (i.e., dolutegravir or darunavir), and did not take into account modifications of the regimen involving the backbone. As a consequence, potential toxicities related to antiretroviral drugs included in the backbone should not have influenced TD rates.

Some limitations should also be noted when interpreting the results of our study. Since this was an observational study from a real-life setting, patients were not randomly assigned to dolutegravir or darunavir, but were prescribed a specific treatment option based on their treating physicians’ preference. The two treatment groups were quite homogeneous for the main characteristics at baseline, but higher rates of HCV coinfected patients and a greater use of tenofovir/emtricitabine as backbone were observed in the darunavir group. These discrepancies could have introduced unmeasured or selection biases, potentially influencing our results despite adjusted analyses. Moreover, the availability of drugs coformulated in single-tablet regimens could also have influenced discontinuation rates. However, this information was not available for all patients in our database. Finally, no data on the use of bictegravir were available in our cohort, and this drug could have valuable characteristics in advanced naïve patients, since it is an integrase inhibitor with a high genetic barrier and is coformulated with an NRTI backbone in a single-tablet regimen. Recent data suggest that bictegravir could have good efficacy and provide durability data in this setting [31,32,50]. Thus, further studies should be performed to determine whether our findings could also be translated to this InSTI.

## 5. Conclusions

Dolutegravir- and darunavir-based regimens showed similar virological efficacy in first-line ART in AIDS- or late-presenter naïve patients. However, there was a higher risk of treatment discontinuation in the first 12 months with dolutegravir, mainly driven by CNS toxicity. Instead, there was a higher risk of TD with darunavir after the first 12 months, mainly driven by treatment simplification and drug–drug interactions. Although more comparative data are needed concerning bictegravir use in this population, the use of this drug in this setting seems promising on the basis of its pharmaceutical characteristics and the data reported in the literature.

## Figures and Tables

**Figure 1 viruses-15-01123-f001:**
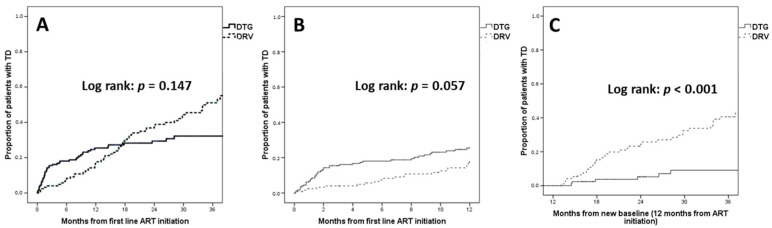
Kaplan–Meier estimates of treatment discontinuation (TD). (**A**) TD over the entire follow-up; (**B**) TD when censoring follow-up at 12 months; (**C**) TD when considering only patients still on first-line treatment at 12 months. Abbreviations: ART, antiretroviral therapy; DRV, darunavir; DTG, dolutegravir; TD, treatment discontinuation.

**Figure 2 viruses-15-01123-f002:**
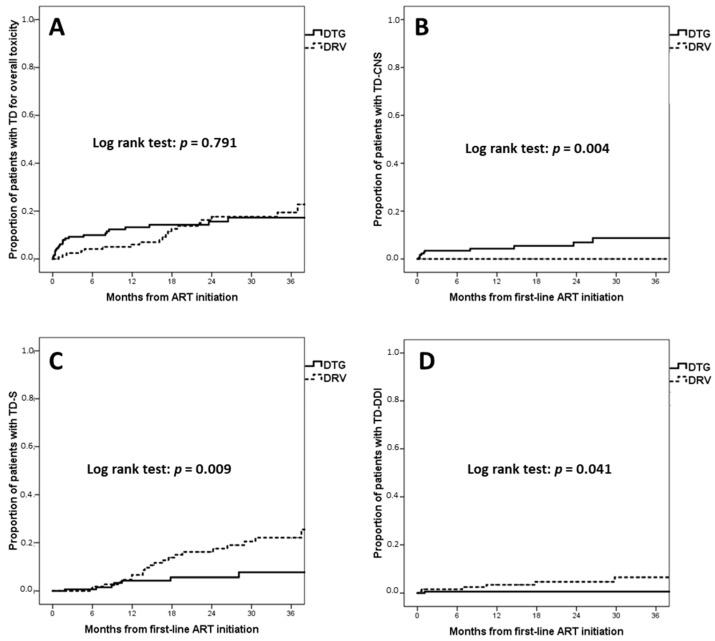
Kaplan–Meier estimates of time to treatment discontinuation (TD) for different reasons. (**A**) TD for overall toxicity; (**B**) TD for central nervous system toxicity; (**C**) TD for simplification; (**D**) TD for drug–drug interactions. Abbreviations: ART, antiretroviral therapy; TD, treatment discontinuation; TD-CNS, TD for central nervous system toxicity; TD-DDI, TD for drug–drug interactions; TD-S, TD for simplification.

**Figure 3 viruses-15-01123-f003:**
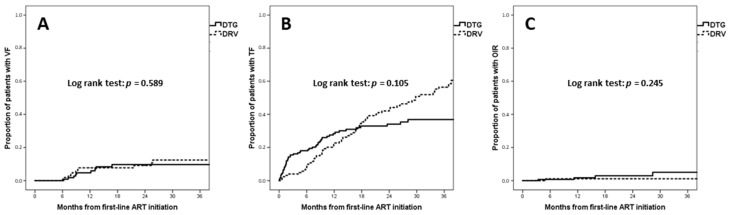
Kaplan–Meier estimates of time to virological failure (**A**), treatment failure (**B**), and optimal immunological recovery (**C**). Abbreviations: ART, antiretroviral therapy; OIR, optimal immunological recovery; TF, treatment failure; VF, virological failure.

**Table 1 viruses-15-01123-t001:** Population’s characteristics at baseline (i.e., date of antiretroviral therapy initiation).

Population Characteristics	Total Population (n = 308)	DTG (n = 181)	DRV (n = 127)	*p*
Age (years)	42.9 (35.1–51.2)	42.5 (35–51)	43.6 (35.2–51.3)	0.601
Male gender	244 (79.2%)	146 (80.7%)	98 (77.2%)	0.456
Caucasian	248 (80.5%)	140 (77.3%)	108 (85%)	0.093
Risk factors:				0.264
Heterosexual	140 (45.5%)	77 (42.5%)	63 (49.6%)	
MSM	113 (36.7%)	67 (37%)	46 (36.2%)	
IDU	10 (3.2%)	5 (2.8%)	5 (3.9%)	
Other/unknown	45 (14.6%)	32 (17.7%)	13 (10.2%)	
HCV+	14 (4.5%)	6 (3.3%)	8 (6.3%)	0.021
HBsAg	10 (3.2%)	5 (2.8%)	5 (3.9%)	0.372
AIDS presenter	124 (40.3%)	68 (37.6%)	56 (44.1%)	0.250
Median time from HIV diagnosis (months)	0.43 (0.18–0.87)	0.37 (0.17–0.78)	0.50 (0.23–1.07)	0.194
Median HIV-RNA baseline (log copies/mL)	5.29 (4.92–5.78)	5.30 (4.91–5.78)	5.27 (4.91–5.77)	0.422
Median CD4+ baseline (cells/µL)	66 (25–122)	66 (25–119)	66 (25–136)	0.812
Median CD4:CD8 ratio	0.10 (0.05–0.20)	0.10 (0.05–0.19)	0.11 (0.05–0.20)	0.568
Backbone:				0.002
TAF/FTC or TDF/FTC	240 (77.9%)	131 (72.4%)	109 (85.8%)	
ABC/3TC	66 (21.4%)	50 (27.6%)	16 (12.6%)	

Abbreviations: 3TC, lamivudine; ABC, abacavir; DRV, darunavir; DTG, dolutegravir; FTC, emtricitabine; HCV, hepatitis C virus; IDU, injecting drug users; MSM; men who have sex with men; TAF, tenofovir alafenamide; TDF, tenofovir disoproxil fumarate.

**Table 2 viruses-15-01123-t002:** Reasons for treatment discontinuation.

Reason	Overall (n = 113/308, 36.7%)	DTG (n = 49/181, 27.1%)	DRV (n = 64/127, 50.4%)
Virological failure	5 (1.6%)	3 (1.7%)	2 (1.6%)
Toxicity, any cause	44 (14.3%)	24 (13.3%)	20 (15.7%)
Gastrointestinal	9 (2.9%)	5 (2.8%)	4 (3.1%)
CNS	10 (3.2%)	10 (5.5%)	0
Simplification	35 (11.4%)	9 (5%)	26 (20.5%)
Intensification	5 (1.6%)	2 (1.1%)	3 (2.4%)
Drug–drug interactions	7 (2.3%)	1 (0.6%)	6 (4.7%)
Other	10 (3.2%)	6 (3.3%)	4 (3.1%)
Death	7 (2.3%)	4 (2.2%)	3 (2.4%)

Abbreviations: CNS, central nervous system; DRV, darunavir; DTG, dolutegravir.

## Data Availability

Recorded data were anonymized and managed according to Good Clinical Practice.

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
