# Peer review of "Efficacy and Durability of Dolutegravir- or Darunavir-Based Regimens in ART-Naïve AIDS- or Late-Presenting HIV-Infected Patients"

_viruses, 2023, doi:10.3390/v15051123_

Round 1

Reviewer 1 Report

authors present retrospective study to compare dolutegravir or darunavir regimens for late initiated treatment of HIV individuals.  Well written and thorough discussion, including limitation of study. Important study, well designed.

Reviewer 2 Report

The manuscript by Fabbiani et al reports treatment findings of dolutegravir- or darunavir-based

regimens in ART-naïve AIDS- or late-presenter HIV-infected patients. The study is based on a retrospective information from multicenter that include ART-naïve adult HIV-1 infected, AIDS- or late-presenter. These patients enrolled in starting first line therapy with dolutegravir or ritonavir/cobicistat-boosted darunavir once daily + two nucleoside reverse transcriptase inhibitors (NRTI) from January 2009 to June 2019.  Therefore, methodology involves data analysis collected electronic data and chart review with no complex experimental methodology. The Authors describe the need for such report is due to the limited availability reports in regarding treatments in ART-naïve AIDS- or late-presenter HIV-infected patients.  However, they don’t describe adequately how novel is their studies as compared to the claimed limited previous studies convincingly describing a justifying for this report that warrants publication.  The writing contains many awkward and long sentences and at times difficult to understand.  This is evident in the Abstract section and throughout the main text.  

-What are the controls.

-With the TD of dolutegravir or darunavir what is the impact of the other drugs.

The writing contains many awkward and long sentences and at times difficult to understand.  This is evident in the Abstract section and throughout the main text.

Reviewer 3 Report

this is a well written study, good english, clearly laid out, that does a retrospective analysis of those with hiv and advanced disease who were treated with dolutegravir or darunavir containing regimes.

the incidence of cns effects with dolutegravir is higher in this population than with other populations, and it is worth explaining this.  ie advanced hiv may have more cns problems baseline and this may affect tolerability of dolutegravir. possibly it would be good to highlight

Round 2

Reviewer 2 Report

The authors addressed the concerns adequately.

Minor editing is required.